# Relations of Conspicuous Consumption Tendency, Self-Expression Satisfaction, and SNS Use Satisfaction of Gen Z through SNS Activities

**DOI:** 10.3390/ijerph182211979

**Published:** 2021-11-15

**Authors:** Seung-A. Shin, Jong-Oh Jang, Jong-Kul Kim, Eun-Hyung Cho

**Affiliations:** 1Department of Sports and Leisure Industry, Sehan University, Dangjin 31746, Korea; ssa0618@sehan.ac.kr; 2Department of Sport Science, Yongin University, Yongin 17092, Korea; 2340505@yongin.ac.kr; 3FA Sports Co., Ltd. 3, Seoul 04998, Korea; t01052416956@gmail.com; 4Korea Institute of Sport Science, Seoul 01794, Korea

**Keywords:** Generation Z, conspicuous consumption propensity, self-expression satisfaction, SNS using satisfaction

## Abstract

The purpose of this study was to analyze the relations of conspicuous consumption tendency, self-expression satisfaction, and SNS use satisfaction of Gen Z through SNS activities. For a week from 17–23 March 2021, an online survey was conducted targeting the enrolled students of university in Chungcheongnam-do. Out of a total of 398 questionnaires, a total of 394 questionnaires (98.9%) were used for the final analysis after excluding four questionnaires with low reliability. This study was analyzed using SPSS by IBM 23.0(New York, NY, USA) and AMOS 21.0 (New York, NY, USA). In the results of this study, first, the factors such as imported goods/famous brands, high-priced articles, and status symbol except for pursuit of trend of conspicuous consumption tendency had significant effects on the self-expression satisfaction. Second, the factors such as imported goods/famous brands, high-priced articles, and status symbol except for pursuit of trend of conspicuous consumption tendency of Gen Z through SNS activities had significant effects on the SNS use satisfaction. Third, the self-expression satisfaction of Gen Z through SNS had significant effects on the SNS use satisfaction.

## 1. Introduction

### 1.1. Necessity of Research

Because modern society develops rapidly, it is known that the generation gap widens, showing a clear difference between generations [1,2]. The Gen Z generation (Gen Z) which shares many characteristics with the Millennials also has its own new behavioral pattern. Today, companies should understand those unique characteristics of Gen Z [3]. Gen Z means the generation born after 1995 [4,5]. The Millennials view that joining a large company (Samsung Electronics, Google, Disney, Apple, etc.) admired by everyone as a ‘dream job’ is a way to promote their success and personal growth [6,7]. On the other hand, Gen Z is not passionate about the hard-earned income after investing much time of their lives in a company, but focusing on their own freedom and interes. In other words, Gen Z longs for creating high profits by freely working in the field they are interested in and confident about, instead of hard-earned income. As a perfect online generation, the whole life of Gen Z is deeply related to the internet. Recently, they have been found to dominate the consumption market through their own methods by using YouTube, online shopping, Instagram, Facebook, and Weibo. In this way, Gen Z is actively utilizing digital technology for consumption activities [8]. In the pre-stage of a product purchase, they objectively search and collect information about products by using social media or community, and prefer the online/mobile purchase of almost every consumer goods item [9].

They not only purchase goods, but also share everything online [10]. They want their thoughts or emotions to be recognized and empathized by online social relationships. Gen Z perceives online not only as a space for sharing information and relationships with others, but also as a space for expressing themselves [11,12,13]. Those self-expression characteristics of Gen Z are clearly shown in the consumption market. After perceiving Gen Z as a new consumption power, to meet their needs, the online consumption market is taking an active attitude such as developing quick distribution services and personalized bidirectional communication services based on digital technology [14].

Gen Z shows a remarkable consumption pattern focusing on luxury brands in order to show themselves off, and they show off and express themselves differently from the existing generations [15]. In relation to this, a research by Barrera and Ponce (2020) noted that the young consumers would show the snob consumption pattern of purchasing luxury brand goods based on their great interest in the luxury category [16]. Taylor and Strutton (2016) reported that they could compare themselves with others by using Facebook, and there would be a significant correlation between the desire for promoting oneself and the tendency to participate in conspicuous consumption [17]. A research by Krause et al. (2019) also presented the significant correlation between use of Instagram and conspicuous consumption tendency [18]. In this perspective, Gen Z which is called perfect online generation is consuming as a means to show off themselves online, which is utilized as a means to express ‘Who they are?’ to other people. The conspicuous consumption tendency through continuous SNS activities is utilized as a space for expressing oneself, and when feeling satisfied with it, it is led to continuous SNS activities.

In the era when everything (interpersonal relationship, shopping, education, communication, hobby, etc.) could be done online, it is important to understand the conspicuous consumption tendency according to the characteristics of Gen Z representing the online market [19,20,21]. Moreover, it would be necessary to have researches on the influence relation between self-expression satisfaction of online activity and continuous online activity intention. 

Thus, it would be important to analyze the characteristics of Gen Z rising as a new leading role of consumption market, and also to understand a certain approach to increase their consumption. However, most of the existing researches are simply related to the consumption results (satisfaction, behavioral intentions, etc.) of Gen Z [22,23,24,25,26].

Thus, this study aimed to understand the accurate consumption tendency of Gen Z by analyzing the relations of conspicuous consumption tendency, self-expression satisfaction, and SNS use satisfaction of Gen Z through SNS activities. The study also aimed to present the marketing sales strategies for drawing them to the consumption market.

### 1.2. Purpose of This Study

The purpose of this study was to analyze the relation of conspicuous consumption tendency, self-expression satisfaction, and SNS use satisfaction of Gen Z through SNS activities. Through this process, this study purposes to present the efficient marketing sales strategies that could be applied to the actual field, by understanding the consumption tendency and characteristics of Gen Z rising as a leading role of future consumption.

## 2. Methods 

### 2.1. Research Subjects

For a week from 17–23 March 2021, a Google online survey was conducted targeting the enrolled students of university in the Chungnam region. Out of a total of 398 questionnaires, a total of 394 questionnaires (98.9%) were used for the final analysis after excluding four questionnaires with low reliability. The general characteristics of the research subjects are presented in Table 1.

### 2.2. Research Tools

#### 2.2.1. Composition of Research Tools

For the questionnaire items related to conspicuous consumption tendency, this study used a preceding research by Jang and Heo (2009) that developed a scale for measuring the conspicuous consumption tendency of adolescent consumers, by modifying/complementing it in a way suitable for this study [27]. For the questionnaire items related to self-expression satisfaction, this study used a research by Park (2006) on the relation between consumer self-expression and satisfaction of online shopping malls, by modifying/complementing it in a way suitable for this study [28]. For the questionnaire items related to SNS use satisfaction, this study used a research by Choi and Park (2019) on the relation between characteristics of SNS fan page and use satisfaction of professional baseball teams [29], by modifying/complementing it in a way suitable for this study. The details are shown in the following in Table 2.

#### 2.2.2. Verification of Validity of Research Tools

To verify the validity and reliability of questionnaire, the construct validity (β) and convergent validity (AVE, CR) were verified by conducting a confirmatory factor analysis. The construct validity is judged to be secured when the β value of observation variable is 0.5 or higher. As an index for understanding the reliability and validity of latent variable itself, the convergent validity is judged to be secured when the value of Average Variance Extracted (AVE) is 0.5 or higher, and the value of construct reliability (CR) is 0.7 or higher [30]. In the results of verification, except for the items of pursuit of trend 10, status symbol 18, and self-expression 1, all the remaining items met the construct validity and convergent validity. In the results of understanding the goodness of fit of the model realized by using the AMOS 21.0, it was shown that χ^2^ = 546.173, *df* = 214, *p* < 0.001, GFI = 0.892, TLI = 0.94, CFI = 0.95, RMR = 0.048, and RMSEA = 0.063, which met the goodness of fit. The concrete contents are presented in Table 3.

#### 2.2.3. Verification of Reliability of Research Tools

The internal consistency of the research tools used for this study was verified. As a result, it was shown as 0.896 for imported goods/famous brands, 0.711 for high-priced articles, 0.87 for pursuit of trend, 0.792 for status symbol, 0.894 for self-expression satisfaction, and 0.931 for SNS use satisfaction, which was higher than the reference value (0.7) of reliability, thus the internal consistency was verified. The detailed contents are presented in Table 4.

### 2.3. Data Processing Methods

Using the SPSS 23.0, the data analysis was conducted as follows. To understand the characteristics of the research subjects, the frequency analysis was performed. To verify the validity (construct validity, convergent validity) and internal consistency of the research tools, the confirmatory factor analysis and reliability analysis were conducted. Then, to verify the degree of independence and correlation between factors, a correlation analysis was conducted. In order to verify the hypotheses of this study, a multiple regression analysis was performed.

## 3. Results 

### 3.1. Correlation Analysis

In the results of verifying the correlation coefficient value between factors through correlation analysis, the value was shown as *r* = 0.209~0.696 (*p* < 0.01), which showed significantly positive (+) correlation between factors. The correlation analysis is judged to have a problem with multicollinearity when the correlation between factors is the reference value or higher (*r* > 0.8). The high correlation between factors means that it is meaningless to divide those two factors as the correlation between them is so natural. The detailed contents of correlation coefficient value between factors presented in this study are shown in Table 5.

### 3.2. Verification of Hypotheses

The purpose of this study was to analyze the relations of conspicuous consumption tendency, self-expression satisfaction, and SNS use satisfaction of Gen Z through SNS activities. Through this process, this study aimed to present the efficient marketing measures that could be applied to the actual field, by understanding the consumption tendency and characteristics of Gen Z rising as a new leading role of future consumption. The results of analyses for examining this purpose are as follows.

#### 3.2.1. Relation between Conspicuous Consumption Tendency and Self-Expression Satisfaction of Gen Z through SNS Activities

The explanatory power of subfactors of conspicuous consumption tendency on the self-expression satisfaction was 38.3%. In detail, only the factors such as imported goods·famous brands (β = 0.150), high-priced articles (β = 0.195), and status symbol (β = 0.307) had significant effects on the self-expression satisfaction, while the pursuit of trend (β = 0.106) did not have significant effects on it. In the results of verifying the goodness of fit of the model, it was shown as suitable (*F* = 61.993, *p* < 0.001) in Table 6.

#### 3.2.2. Relation between Conspicuous Consumption Tendency and SNS Use Satisfaction of Gen Z through SNS Activities

The explanatory power of subfactors of conspicuous consumption tendency on the SNS use satisfaction was 15.2%. In detail, only the factors like imported goods·famous brands (β = 0.192), high-priced articles (β = 0.159), and status symbol (β = 0.215) had significant effects on the SNS use satisfaction while the pursuit of trend (β = 0.118) did not have significant effects on it. In the results of verifying the goodness of fit of the model, it was shown as suitable (*F* = 18.628, *p* < 0.001) in Table 7.

#### 3.2.3. Relation between Self-Expression Satisfaction and SNS Use Satisfaction of Gen Z through SNS Activities

The explanatory power of self-expression satisfaction on the SNS use satisfaction was 18.2%. In detail, the self-expression satisfaction (β = 0.429) had significant effects on it. In the results of verifying the goodness of fit of the model, it was shown as suitable (*F* = 88.593, *p* < 0.001) in Table 8.

## 4. Discussions

Based on the results drawn for examining the relations of conspicuous consumption tendency, self-expression satisfaction, and SNS use satisfaction of Gen Z through SNS activities, this study aims to have discussions as follows. 

First, in the results of verifying, only the factors such as imported goods·famous brands, high-priced articles, and status symbol except for pursuit of trend of conspicuous consumption tendency of Gen Z through SNS activities had significant effects on the self-expression satisfaction. This result supports the results of existing researches reporting that the consumption tendency of Gen Z does not pursue the trend unconditionally, but pursue each one’s individuality [31,32,33]. Among the researches examining the relation between conspicuous consumption tendency and self-expression satisfaction, a research by Lee (2020) indicated that when the consumers voluntarily reveal the brands they consume on SNS, they are feeling self-satisfaction and are also conscious of the attention from the peer group and acquaintances based on their desire to feel a sense of belonging to a group, which supports the results of this study [34]. As such, the consumers feel satisfied and recognized by others like friends and acquaintances by voluntarily sharing what they consumed in writing and photos on SNS. 

Moreover, a research by Kim and Kim (2018) on the degree of self-expression satisfaction according to conspicuous tendency through Instagram indicated that the consumers who uploaded the review of visits to Seoul luxury hotels through Instagram would feel satisfied by consuming the space of hotels as a type of conspicuous self-expression pretending to live a fancy, wonderful, and rich life to others [35]. A research by Lee and Lee (2013) on the relation between consumption value of luxury brands and consumer happiness emphasizes the necessity of promotion strategies reflecting that the consumers who pursue the conspicuous/social consumption value regard symbolism, prestige, and honor as important, and also feel satisfied through them, which is supporting the results of this study [36]. Additionally, a research by Krause et al. (2019) reported that showing off and revealing one’s desire to get socially recognized through SNS are related to the formation of positive relationships in interests, which accords with the results of this study. As such, Gen Z feels satisfied with their own looks shown to others, by purchasing products to show off their own individuality, rather than purchasing mass-produced products that could be easily purchased. When the Millennials purchase high-priced products (luxury goods, limited edition, etc.), they mostly aim to feel satisfied by purchasing a single great product in the aspect of self-reward. However, Gen Z considers luxury goods as necessities for life, which shows the characteristics of the conspicuous consumption tendency of Gen Z. 

Second, in the results of verifying, only the factors such as imported goods·famous brands, high-priced articles, and status symbol except for pursuit of trend of conspicuous consumption tendency of Gen Z through SNS activities had significant effects on the SNS use satisfaction. In relation to this result, a research by Lee and Kim (2020) reported that the consumers who performed a conspicuous consumption behavior and then uploaded it on Instagram, feel positive emotions, and such emotions have significant effects on the consumer happiness (satisfaction) [37], which is supporting the results of this study. Moreover, a study by Kim (2018) on the relation between conspicuous consumption tendency by sensitivity to honor and self-esteem, and college life satisfaction, the consumers with strong sensitivity to honor have a strong tendency to exaggeratedly reveal more than what they have such as purchase of product, travel, eating out, and leisure activity through various SNS such as Instagram, Twitter, and Facebook, in order not to lose honor, or to raise it [38]. In other words, it reports that when the consumers feel satisfied with such activities, it has effects on their continuous consumption behavior, which is supporting the results of this study. A research by Widjajanta et al. (2018) on the relation between social media use and consumption pride (satisfaction) showed that the self-display through the use of social media would increase their own pride [39]. In other words, revealing and showing off oneself by sharing the consumption through social media could have positive effects on the increase of psychological satisfaction. Additionally, according to Ferreira (2016), the consumers ostentatiously show consumption behaviors to reveal their own social status; the social media provides a new platform to reveal this conspicuous consumption of individuals; and there is a positive correlation between social media use and conspicuous consumption, which explains the relation between conspicuous consumption and SNS use satisfaction [40]. This way, the SNS is usually used for the formation of mutual relationships and sharing information through the internet; thus, the consumers have strong desires to show themselves off and to be recognized in relationships through SNS. When such desires are met through the SNS, the consumers feel satisfied with the SNS use. 

Third, in the results of verifying, the self-expression satisfaction of Gen Z through SNS had significant effects on the SNS use satisfaction. This result accords with the results of a research by Lee (2020) reporting that the active self-expression through SNS could improve the SNS use satisfaction and also have positive effects on the life satisfaction [41]. A research by Kim, Baek, and Choo (2017) also reported that the high self-expression desire through the SNS could be led to the tendency to show active SNS activities such as taking selfies and sharing daily life. Additionally [42], a research by Shane-Simpson, Manago, Gaggi, and Gillespie-Lynch (2018) shows that the Instagram users are more active in self-expression by showing higher-level of self-disclosure than the Facebook users, which accords with the results of this study [43]. Moreover, a research by DeAndrea, Shaw, and Levine (2010) reported that the cultural formation of SNS could have positive effects on the self-construal and self-expression of consumers, and moreover, it would form a positive attitude in user satisfaction, which is supporting the results of this study [44]. A research by Park (2016) revealed that the satisfaction of social game use becomes higher in the case when the self-expression is actively expressed within a social game [45]. Furthermore, a research by Park, Yoon, and Kim (2015) reports that the satisfaction becomes increased in the case when the self-expression is actively performed within tourism social media, and moreover, this is led to its continuous use, which is supporting the results of this study [46]. Therefore, as a perfect online generation, Gen Z wants to get empathized by sharing their own daily life, individuality, and self-expression with many people beyond the formation of limited social relationships such as offline acquaintances and colleagues.

## 5. Conclusions and Suggestions

### 5.1. Conclusions

The purpose of this study was to analyze the relations of conspicuous consumption tendency, self-expression satisfaction, and SNS use satisfaction of Gen Z through SNS activities. Through this process, this study aimed to present the efficient marketing measures that could be applied to the actual field, by understanding the consumption tendency and characteristics of Gen Z rising as a new leading role of future consumption. Based on such results, this study aimed to put together conclusions as follows. 

First, only the factors like imported goods·famous brands, high-priced articles, and status symbol except for pursuit of trend of conspicuous consumption tendency had significant effects on the self-expression satisfaction. Second, only the factors like imported goods·famous brands, high-priced articles, and status symbol except for pursuit of trend of conspicuous consumption tendency of Gen Z through SNS activities had significant effects on the SNS use satisfaction. Third, the self-expression satisfaction of Gen Z through SNS had significant effects on the SNS use satisfaction. 

Based on the above, as a rising leading role of new consumption, Gen Z leads the popularization of luxury goods, sensitively responds to their own status and individuality, and feels satisfied with the formation of social relationships by using it. Thus, Gen Z is a new generation who will lead the future consumption trend in various industrial areas; hence, the domestic/foreign companies would need to have efficient marketing and sales strategies by applying the expression of consumers’ individuality, diversification of option, high-quality strategy compared to price, VIP system, and personal signature design products such as BESPOKE of Samsung Electronics, color diversification of GALAXY, and limited edition of collaboration of a famous sports brand. 

### 5.2. Suggestions

For follow-up researches, this study aims to make suggestions as follows. First, this study conducted the research by targeting the enrolled students of university in the Chungnam region, thus it is limited to generalize the results of this study. Thus, by conducting the analysis on the consumption tendency by expanding the sample from university students to the whole Gen Z, it would be possible to accurately understand the characteristics of Gen Z, and also to establish the marketing strategies meeting their needs. Second, as shown in this study, it would be necessary to conduct a research on the characteristics and conspicuous consumption tendency of a group that is more actively utilizing SNS activities, which could draw data that would be helpful for planning the marketing strategies through SNS. Third, it would also be needed to examine the relations of various factors for analyzing the current market condition by reflecting the characteristics of Gen Z besides the factors of this study, which could draw some analysis data in a more multilateral perspective. Fourth, in future research, it will be necessary to use SEM to take errors into account.

## Figures and Tables

**Table 1 ijerph-18-11979-t001:** General characteristics (N = 394).

	Classification	Frequency	%
Gender	Male students	232	58.9
	Female students	162	41.1
School year	1st	155	39.3
	2nd	100	25.4
	3rd	86	21.8
	3rd	86	21.8
	4th	53	13.5
Period of SNS use	Less than 1 year	19	4.8
	Less than 2 years	10	2.5
	Less than 3 years	19	4.8
	Less than 4 years	27	6.9
	5 years or up	319	81.0
The number of weekly SNS upload	1 time	244	61.9
	2 times	49	12.4
	3 times	35	8.9
	4 times	13	3.3
	5 times	6	1.5
	6 times or more	47	11.9
The number of daily comment/DM/communication	1 time	145	36.8
	2 times	38	9.6
	3 times	49	12.4
	4 times	22	5.6
	5 times	13	3.3
	6 times or more	127	32.2
	Total	394	100

**Table 2 ijerph-18-11979-t002:** Composition of questionnaire.

Factor	Contents	No. of Items
Demographic characteristics	Gender, school year, period of SNS use, the number of weekly SNS upload, the number of daily comment/DM/communication	5
Conspicuous consumption tendency	Conspicuous factor of imported goods/famous brands(6) Conspicuous factor of high-priced articles(3) Conspicuous factor of pursuit of trend(5) Conspicuous factor of status symbol(4)	18
Self-expression satisfaction	Self-expression satisfaction	4
SNS use satisfaction	SNS use satisfaction	4
	Total	31

**Table 3 ijerph-18-11979-t003:** Verification of validity.

Item	β	B	S.E.	AVE	CR
Imported goods·famous brands1	0.635	1	-	0.605	0.901
Imported goods·famous brands2	0.808	1.181	0.078		
Imported goods·famous brands3	0.757	1.194	0.094		
Imported goods·famous brands4	0.821	1.154	0.086		
Imported goods·famous brands5	0.821	1.238	0.092		
Imported goods·famous brands6	0.872	1.277	0.091		
High-priced articles7	0.862	1	-	0.620	0.829
High-priced articles8	0.915	1.045	0.049		
High-priced articles9	0.707	0.879	0.056		
Pursuit of trend11	0.795	1	-	0.671	0.891
Pursuit of trend12	0.816	1.055	0.06		
Pursuit of trend13	0.817	1.056	0.06		
Pursuit of trend14	0.832	1.183	0.066		
Status symbol15	0.758	1	-	0.507	0.755
Status symbol16	0.802	1.045	0.07		
Status symbol17	0.684	0.924	0.072		
Self-expression satisfaction2	0.832	1	-	0.686	0.868
Self-expression satisfaction3	0.861	1.097	0.054		
Self-expression satisfaction4	0.885	1.022	0.049		
SNS use satisfaction1	0.833	1	-	0.764	0.928
SNS use satisfaction2	0.952	1.073	0.042		
SNS use satisfaction3	0.922	1.014	0.042		
SNS use satisfaction4	0.81	0.852	0.044		

Deleted items: Pursuit of trend10, Status symbol18, Self-expression satisfaction1, GFI = 0.892, TLI = 0.94, CFI = 0.95, RMR = 0.048, RMSEA = 0.063, χ^2^ = 546.173, *df* = 214, *p* < 0.001.

**Table 4 ijerph-18-11979-t004:** Verification of reliability.

Factor	Subfactor	Cronbach’s α
Conspicuous consumption tendency	Imported goods·famous brands	0.896
	High-priced articles	0.711
	Pursuit of trend	0.870
	Status symbol	0.792
Self-expression satisfaction	Self-expression satisfaction	0.894
SNS use satisfaction	SNS use satisfaction	0.931

**Table 5 ijerph-18-11979-t005:** Verification of correlation.

	1	2	3	4	5	6
Imported goods·famous brands	1					
High-priced articles	0.512 **	1				
Pursuit of trend	0.696 **	0.410 **	1			
Status symbol	0.640 **	0.467 **	0.597 **	1		
Self-expression satisfaction	0.520 **	0.459 **	0.474 **	0.558 **	1	
SNS use satisfaction	0.329 **	0.309 **	0.209 **	0.341 **	0.429 **	1

** *p* < 0.01.

**Table 6 ijerph-18-11979-t006:** Relation between conspicuous consumption tendency and self-expression satisfaction of Gen Z through SNS activities.

Dependent	Independent	β	S.E.	t	*p*
Self-expression satisfaction	(Constant)	-	0.121	4.332	0.001
	Imported goods·famous brands	0.150	0.075	2.422	0.016
	High-priced articles	0.195	0.048	4.135	0.001
	Pursuit of trend	0.106	0.069	1.848	0.065
	Status symbol	0.307	0.057	5.617	0.001

*R*^2^ = 0.389, adj *R*^2^ = 0.383, *F* = 61.993, *p* < 0.001.

**Table 7 ijerph-18-11979-t007:** Relation between conspicuous consumption tendency and SNS use satisfaction of Gen Z through SNS activities.

Dependent	Independent	β	S.E.	t	*p*
SNS use satisfaction	(Constant)	-	0.130	16.803	0.001
	Imported goods·famous brands	0.192	0.081	2.640	0.009
	High-priced articles	0.159	0.051	2.871	0.004
	Pursuit of trend	0.118	0.074	1.754	0.080
	Status symbol	0.215	0.061	3.350	0.001

*R*^2^ = 0.161, adj *R*^2^ = 0.152, *F* = 18.628, *p* < 0.001.

**Table 8 ijerph-18-11979-t008:** Relation between self-expression satisfaction and SNS use satisfaction of Gen Z through SNS.

Dependent	Independent	β	S.E.	t	*p*
SNS use satisfaction	(Constant)	-	0.106	21.496	0.001
	Self-expression satisfaction	0.429	0.042	9.412	0.001

*R*^2^ = 0.184, adj *R*^2^ = 0.182, *F* = 88.593, *p* < 0.001.

## Data Availability

The data presented in this study are available on request from the corresponding author. The data are not publicly available due to privacy issues.

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
