# Peer review of "Relations of Conspicuous Consumption Tendency, Self-Expression Satisfaction, and SNS Use Satisfaction of Gen Z through SNS Activities"

_ijerph, 2021, doi:10.3390/ijerph182211979_

Round 1

Reviewer 1 Report

Title and abstract:
“Relations of Conspicuous Consumption Tendency, Self-Expression Satisfaction, and SNS Use Satisfaction of Gen Z through SNS Activities”
The topic is interesting, the title is informative, and catchy and could express the tone of writing that also includes key variables. 
The abstract could successfully convey the main topics of the study while highlighting the importance of the research. Besides, it was concise and attract readers. Please revise the statement “Using the SPSS 23.0 and AMOS 21.0, this study conducted the frequency analysis, confirmatory factor analysis, reliability analysis, correlation analysis, and multiple regression analysis”. The author does not need to mention details of SEM analysis. Also, research findings are addressed. 
Introduction: 
Unfortunately, key terms are not well-explained and introduced to readers. The other observed flaw is related to the missing citations. For example, on page 2 lines 50-51 “They not only purchase goods but also share everything online. They want their thoughts or emotions to be recognized and empathized by online social relationships.” The author should once again read through the introduction and fix this issue. 
Literature review and hypothesis development:  
The paper should incorporate a more solid argumentation that allows justifying the reason that allows selecting the variables that are considered in the analysis. Furthermore, as far as possible, the theoretical framework should be sufficiently solid to justify that the relevant variables that should explain the phenomenon under study are those considered in the analysis and only those variables. Unfortunately, there is no solid theory is provided which makes further consideration for publication difficult. Improve the argumentation of the hypothesis.
Method and results: 
More information needs to be provided on how the data were collected when and how the data were collected? And y such sample could be a proper representative for such study? The methodology used should be introduced and the author should elaborate more on the method and its suitability and adaptability! 
Discussion and conclusion: 
The quality of discussion and conclusion is somehow acceptable. 
In the discussion section, you have to Relate your findings to those of previous studies, for example, whether your results support or deviate from results in previous studies. Explain how the study adds to previous knowledge. Remember to mention any possible alternative explanations for the results. 

Author Response

Here are some comments.

  1. title of sub-introduction

1.1. necessities of research

‘necessities of research’ would be not an ideal title.

Edited the content.

  1. Literature review

Before presenting hypotheses, the authors should provide conceptual and empirical evidence from previous studies, however, the literature view section is not provided.

Edited the content.

  1. The definition of the concepts

The authors did not present definitions of constructs in the proposed model. Moreover, the relationship between antecedents and outcomes should be discussed.

Edited the content.

  1. Demographic information

School year – third year is not missing…

Edited the content.

  1. Results

The authors present two reliability results such as CR and Cronbach’s alpha. Some constructs show very different gaps. What do you think about the results?

It is judged that the meaning of Cronbach's alpha value for confirming internal consistency and the concept reliability of the concept for confirming the focus validity of the research tool are different. Both are presented.

  1. Analysis

Why did authors use AMOS but not use SEM?

Thanks for the reviewer's advice. SEM is considered appropriate in that it considers the error of the measurement variable. However, the model fit of the research model was not correct, so I had no choice but to give up. This will be supplemented in a follow-up study.

  1. Analysis

If your model is to exam the relationships between conspicuous consumption, self-expression satisfaction mediated by SNS use satisfaction., why did the authors did not provide the mediation test results?

I understand and appreciate the reviewers' opinions. However, the purpose of this study is not to focus on mediator variables, but rather to study whether independent variables affect various variables. In a follow-up study, we will proceed with a study that complements this.

  1. Discussion

In discussion, the results should be discussed and provide theoretical and practical implications, however, I have more information about summarized findings of previous research.

Previous studies on the conspicuous consumption tendency of Generation Z are lacking. Nevertheless, i tried to present similar papers as evidence as much as possible. In a follow-up study, i will supplement the limitations of the current study by conducting additional research related to this. i hope you understand these efforts.

  1. References

Please check the references out again. The format is not consistent. Especially, authors presented full names of some Korean authors’ articles.

Edited the content.

Thank you once again for reviewing this thesis with your precious time. I would also like to thank you for your help in making this thesis high-quality.

Reviewer 2 Report

The introduction and literature should exemplify the topic addressed, describe the main specific contributions made to date, what are the research gaps that the research is attempting to fill, how previous contributions relate to the contribution that is intended to be made in this paper and, if it is the case, who previously suggested the need to make the analysis included in this new study. The theoretical foundation on which the model is based should be strengthened, and not only mentioned in the introduction. The authors should take as a starting-point one or more sufficiently contrasted theories and apply them to this new context of analysis to justify the need to develop this new research. The paper should incorporate a more solid argumentation that allows justification of the reason for the selection of the explanatory variables that are considered in the empirical analysis. Furthermore, as far as possible, the theoretical framework should be sufficiently solid to justify that the relevant variables that should explain the phenomenon under study are those considered in the analysis and only those variables. In short, why is this research necessary; why are you using the theoretical model; what are the research gaps; and what recent justification have you provided for the aforementioned?

No justification was given to the hypotheses, so recent academic literature sources should be used to adequate justify the hypotheses. The theoretical/conceptual framework was also not discussed - a full discussion of the framework is required to justify your research. Furthermore, you need to consult a much wider array of sources to justify your hypotheses and expand your literature review in order to create the golden thread.

Please include and use separate headings to show both the theoretical contributions and practical implications sections

Overall, the topic is very relevant, but the structure of the article is also not adequate in terms of the flow and its current format (perhaps rearrange some sections and add more subheadings to create the golden thread) and there also were a number of grammatical errors, so this paper is in need of some more editing by a professional language editor.

Author Response

(The authors gave the same response as above.)

Reviewer 3 Report

Thank you for giving me this opportunity to review this manuscript. 

This study examined the relationship between conspicuous tendency, self-expression satisfaction, and SNS use satisfaction among Generation Z. 

Here are some comments.

1. title of sub-introduction

1.1. necessities of research

‘necessities of research’ would be not an ideal title.

2. Literature review

Before presenting hypotheses, the authors should provide conceptual and empirical evidence from previous studies, however, the literature view section is not provided.

3. The definition of the concepts

The authors did not present definitions of constructs in the proposed model. Moreover, the relationship between antecedents and outcomes should be discussed.

4. Demographic information

School year – third year is not missing…

5. Results

The authors present two reliability results such as CR and Cronbach’s alpha. Some constructs show very different gaps. What do you think about the results?

6. Analysis

Why did authors use AMOS but not use SEM?

7. Analysis

If your model is to exam the relationships between conspicuous consumption, self-expression satisfaction mediated by SNS use satisfaction., why did the authors did not provide the mediation test results?

8. Discussion

In discussion, the results should be discussed and provide theoretical and practical implications, however, I have more information about summarized findings of previous research.

9. References

Please check the references out again. The format is not consistent. Especially, authors presented full names of some Korean authors’ articles.

Author Response

(The authors gave the same response as above.)

Round 2

Reviewer 2 Report

This comment has not been suitably addressed: The introduction and literature should exemplify the topic addressed, describe the main specific contributions made to date, what are the research gaps that the research is attempting to fill, how previous contributions relate to the contribution that is intended to be made in this paper and, if it is the case, who previously suggested the need to make the analysis included in this new study. The theoretical foundation on which the model is based should be strengthened, and not only mentioned in the introduction. The authors should take as a starting-point one or more sufficiently contrasted theories and apply them to this new context of analysis to justify the need to develop this new research. The paper should incorporate a more solid argumentation that allows justification of the reason for the selection of the explanatory variables that are considered in the empirical analysis. Furthermore, as far as possible, the theoretical framework should be sufficiently solid to justify that the relevant variables that should explain the phenomenon under study are those considered in the analysis and only those variables. In short, why is this research necessary; why are you using the theoretical model; what are the research gaps; and what recent justification have you provided for the aforementioned?

There is still NO justification was given to the hypotheses, though some recent academic literature sources were in the introduction, but was not used in a separate sections to justify the hypotheses. The theoretical/conceptual framework was also not discussed - a full discussion of the framework is required to justify your research. Furthermore, you need to consult a much wider array of sources to justify your hypotheses and expand your literature review in order to create the golden thread.

This comment was also ignored: Please include and use separate headings to show both the theoretical contributions and practical implications sections

Overall, the topic is very relevant, but the structure of the article is also not adequate in terms of the flow and its current format (perhaps rearrange some sections and add more subheadings to create the golden thread) and there also were a number of grammatical errors, so this paper is in need of some more editing by a professional language editor.

Author Response

As the reviewer said, the reason why this thesis differs from previous studies was written in the necessity section.

First, it emphasized the need to understand the characteristics of Generation Z.

Second, the characteristics of the recent generation Z were written.

Third, how the existing generation and Generation Z differ from each other is presented.

Fourth, it emphasized the need to understand the consumption propensity of Generation Z online.

Fifth, the relationship between existing online activities and self-expression satisfaction was written.

Sixth, in spite of the diverse consumption characteristics of Generation Z, the existing studies wrote the problem that only consumption results were prepared.

These six points differentiate this thesis from existing thesis.

Online characteristics of Generation Z, conspicuous consumption tendency, self-expression satisfaction, and SNS use satisfaction were selected in the necessity section.

  1. The hypothesis was deleted.

Thank you for your valuable time for our good review.

Thank you for revising the manuscript. There is not sufficient information about discriminant validity because of high correlation between constructs. Moreover. this study explored the proposal model by using multiple regressions so the generalization of this study would be a limitation. This authors reported this issue in their limitations and future research suggestions.

Thank you for your valuable time for our good review.

Reviewer 3 Report

Thank you for revising the manuscript. There is not sufficient information about discriminant validity because of high correlation between constructs. Moreover. this study explored the proposal model by using multiple regressions so the generalization of this study would be a limitation. This authors reported this issue in their limitations and future research suggestions. 

Author Response

It would be important to analyze the characteristics of Gen Z rising as a new leading role of consumption market, and also to understand a certain approach to in-crease their consumption. However, most of the existing researches are simply related to the consumption results(satisfaction, behavioral intentions, etc.) of Gen Z(Davies, 2020; Kirvesmies, 2018; Puiu, 2016; Tunsakul, 2020; Wansi, 2020).

As shown in the main text, it is important to analyze the characteristics of the current generation Z and understand the approach, and present previous studies that mainly focused on research satisfaction and behavioral intentions.

Thank you for your valuable time for our good review.